

# Monitoring the training dose and acute fatigue response during elbow flexor resistance training using a custom-made resistance band

Jingjing Yang[1], Hongbin Xu[2], Juke Liang[2], Jongyeob Jeong[3] and Taojin Xu[2,3]

[1] Faculty of Civil Aviation and Aeroautics, Kunming University of Science and Technology, Kunming, China
[2] College of Mechanical Engineering, Chongqing University of Technology, Chongqing, China
[3] Graduate School of Sciences and Technology for Innovation, Yamaguchi University, Ube, Japan

Corresponding author
Taojin Xu,
w503wc@yamaguchi-u.ac.jp

## ABSTRACT

**Background:** Home-based resistance training offers an alternative to traditional, hospital-based or rehabilitation center-based resistance training and has attracted much attention recently. However, without the supervision of a therapist or the assistance of an exercise monitoring system, one of the biggest challenges of home-based resistance training is that the therapist may not know if the patient has performed the exercise as prescribed. A lack of objective measurements limits the ability of researchers to evaluate the outcome of exercise interventions and choose suitable training doses.

**Objective:** To create an automated and objective method for segmenting resistance force data into contraction phase-specific segments and calculate the repetition number and time-under-tension (TUT) during elbow flexor resistance training. A pilot study was conducted to evaluate the performance of the segmentation algorithm and to show the capability of the system in monitoring the compliance of patients to a prescribed training program in a practical resistance training setting.

**Methods:** Six subjects (three male and three female) volunteered to participate in a fatigue and recovery experiment (5 min intermittent submaximal contraction (ISC); 1 min rest; 2 min ISC). A custom-made resistance band was used to help subjects perform biceps curl resistance exercises and the resistance was recorded through a load cell. The maximum and minimum values of the force-derivative were obtained as distinguishing features and a segmentation algorithm was proposed to divide the biceps curl cycle into concentric, eccentric and isometric contraction, and rest phases. Two assessors, who were unfamiliar with the study, were recruited to manually pick the visually observed cut-off point between two contraction phases and the TUT was calculated and compared to evaluate performance of the segmentation algorithm.

**Results:** The segmentation algorithm was programmatically implemented and the repetition number and contraction-phase specific TUT were calculated. During isometric, the average TUT ($3.75 \pm 0.62$ s) was longer than the prescribed 3 s, indicating that most subjects did not perform the exercise as prescribed. There was a

good TUT agreement and contraction segment agreement between the proposed algorithm and the assessors.

**Conclusion:** The good agreement in TUT between the proposed algorithm and the assessors indicates that the proposed algorithm can correctly segment the contraction into contraction phase-specific parts, thereby providing clinicians and researchers with an automated and objective method for quantifying home-based elbow flexor resistance training. The instrument is easy to use and cheap, and the segmentation algorithm is programmatically implemented, indicating good application prospect of the method in a practical setting.

# INTRODUCTION

Aging and illness can cause muscle wasting and weakness, eventually limiting the physical function of limbs (*Liu & Latham, 2009*). Resistance training is an effective physical rehabilitation method used to improve muscle strength and physical function in elderly and patients with disability, recommended by national health organizations, such as the American College of Sports Medicine and the American Heart Association (*Pollock et al., 1998*; *American College of Sports Medicine, 2009*). The population in need of physical rehabilitation is constantly increasing, however individuals, especially older adults and those living in rural areas (*Schutzer & Graves, 2004*; *Thiebaud, Funk & Abe, 2014*), lack easy access to public rehabilitation centers, because of financial or physical constraints or limited therapist availability (*Schutzer & Graves, 2004*; *Far et al., 2015*). Even though the increase in the number of rehabilitation therapists has outpaced the population growth in recent years (*Wilson, Lewis & Murray, 2009*), there is still a shortage of trained rehabilitation service providers in aging societies.

Home-based resistance training offers an alternative to traditional, hospital-based or rehabilitation center-based resistance training and has attracted much attention recently (*Thiebaud, Funk & Abe, 2014*; *Dobkin, 2017*). After the physiotherapist provides the patient with initial instructions on how to perform the exercise, home-based trainees need to complete the entire training program on their own. Without the supervision of a therapist or help of an exercise monitoring system and considering that the training is a long process that lasts weeks or months, one of the biggest challenges in home-based resistance training is that subjects do not follow their exercise prescription (*Rathleff et al., 2015*; *Faber et al., 2015*; *Riel et al., 2018*). In particular, it has been shown that patients perform the exercises either too fast or too slow, resulting in too short or too long contraction time with too few or too many repetitions (*Rathleff et al., 2016*). A study of 29 participants performing shoulder abduction exercises demonstrated that, at follow-up after 2 weeks of unsupervised home-based exercises, less than 25% of the participants followed the instructions for time-under-tension (TUT) and performed correctly the exercise (*Faber et al., 2015*). The main concern for patients not receiving the prescribed exercise dosage is that decisions on further progression or cessation of a specific program

are difficult. The lack of objective measurements limits the ability of clinicians and researchers to evaluate the outcome of exercise interventions. This makes it virtually impossible to ascertain if lack of improvement is due to incorrect exercise, dosage, or poor adherence (*Rathleff et al., 2016*). Therefore, an exercise monitoring system is necessary and useful for home-based trainees to quantify the exercise dosage and most importantly, to improve adherence to the training program.

Regarding the monitoring of resistance training exercise, training details, such as number of sets and repetitions, rest periods, TUT, velocity, and muscle actions are important variables for monitoring (*Scott et al., 2016*), as they are of crucial importance for strength and power adaptation (*Kraemer & Ratamess, 2004*; *Bird, Tarpenning & Marino, 2005*; *Crewther, Cronin & Keogh, 2006*). The number of repetitions and sets, as well as the TUT can be quantified using sensors, such as the BandCizer (BandCizer Aps, Odense, Denmark) (*Riel et al., 2018*) and a stretch sensor (*Kappel et al., 2012*). The BandCizer is a valid tool that can quantify contraction time, the number of repetitions performed and the force used to stretch the elastic band (*Riel et al., 2018*; *Rathleff Michael et al., 2015*). However, the BandCizer uses two paired magnets to measure the deformation of the band and the deformation is susceptible to factors such as sensor placement (*Rathleff Michael et al., 2015*). Furthermore, since different muscle action phases have different training effect on muscles, differentiation between contraction-specific TUT may be important in some cases (*Rathleff Michael et al., 2015*). Dynamic muscular strength improvements were maximal when eccentric (ECC) action was included in the training program (*Kraemer & Ratamess, 2004*; *Komi, Kaneko & Aura, 1987*). Few studies have discussed how to segment sensor data based on muscle actions. Rathleff et al. developed a custom-written MATLAB program to segment stretch-sensor data into concentric (CON), quasi-isometric, and ECC phases and calculated the TUT of each contraction phase (*Faber et al., 2015*; *Rathleff Skovdal, Thorborg & Bandholm, 2013*). However, it should be noted that in their program, the assessor who rated the data needed to manually select the start and end points of each contraction phase, which led to program inefficiency. It required, on average, 55 s to determine total TUT for 10 repetitions from the stretch-sensor recordings and 3–10 min to rate the contraction-phase specific TUT for 10 repetitions (*Rathleff Skovdal, Thorborg & Bandholm, 2013*).

In general, exercise is accompanied by muscle fatigue (*Andrew & Parker, 1979*; *Hughes et al., 1984*). The fatigue and the recovery rate reflect the metabolic capacity of the patient, providing the physiotherapist with an intuitive perception of the subject's exercise capacity. Moreover, fatigue experienced by the subjects may decrease their compliance to exercise programs. Individuals who report high levels of fatigue are especially likely to drop out of an exercise program (*Hughes et al., 1984*). Numerous methods have been proposed to monitor muscular fatigue associated with exercise (*Dong et al., 2014*; *Al-Mulla, Sepulveda & Colley, 2011*) and surface electromyography (sEMG) is the most common approach used (*Dong et al., 2014*; *Nazmi et al., 2016*; *Chang, Liu & Wu, 2012*; *Al-Mulla & Sepulveda, 2010*). The regression slope of the linear regression of median frequency has been used as an important muscle fatigue index (*Chang, Liu & Wu, 2012*). Recently, some fatigue-recovery protocols were deliberately designed to study the

fatigability and the recovery ability of knee extensor muscle under intermittent, isometric (ISOM), or dynamic maximal voluntary contractions in old adults (*Callahan, Foulis & Kent-Braun, 2009*; *Callahan, Umberger & Kent, 2016*; *Sundberg et al., 2018*). Knee extension torque and power were measured using a dynamometer and the fatigability and recovery ability were expressed as reductions and increases in torque, velocity and power.

Considering all these factors, the main objective of this study was to present a simple and automated method for calculating the repetition number and contraction phase-specific TUT during elbow flexor resistance training. Therefore, the biceps curl with a resistance band was selected, since it is a typical single-joint exercise focused on the strength and power improvement of elbow flexor muscles and it is simple and suitable for home-based resistance training. A pilot experiment including six subjects was conducted to evaluate performance of the segmentation algorithm and some fatigue and recovery indexes were proposed to quantify a subject's acute fatigue and recovery response to intermittent submaximal contraction (ISC).

## MATERIALS AND METHODS

### Concepts and big picture

Accommodating resistance devices, such as bands and chains, are useful methods to maximize gains in strength and hypertrophy (*Kraemer & Ratamess, 2004*). A significant feature of these devices is the varying resistance throughout the range of motion, where the level of resistance varies according to the rate and the maximum stretch of the material (greater stretch produces greater resistance) and similar results were obtained when resistance training was performed using traditional equipment (machines and weights) (*Souza et al., 2019*). Accordingly, in this study the biceps curl with a resistance band was selected as the training exercise. Figure 1 illustrates the concepts and flow chart of the system. Firstly, a framework for automatically segmenting force data into small segments based on muscle actions was built. Then, several indexes were introduced to quantify a subject's fatigue and recovery response during ISC.

As illustrated in Fig. 1, the subject sits on a wheelchair while performing the biceps curl exercise against resistance produced by a custom-made resistance band. A load cell is connected to the band to measure the resistance. Analog signals corresponding to the resistance were sampled and digitized by an analog-to-digital (A/D) converter and the data were sent to a desktop computer through an Arduino board. A graphical user interface (GUI) was built for monitoring the training process and the force signals were acquired and saved for further analysis and post-processing. First, a peak detection method was used to find signal valleys, to divide the force data into small segments ($F_i(t)$), then the number of repetitions was calculated. In each segment, the first order derivative of force vs. time was numerically calculated, and the maximum and minimum values were taken as the discriminating feature to segment the training cycle into the CON, ECC and ISOM contraction phases. The average force ($F_{ave}(i)$) during ISOM contraction was calculated and the decreasing slopes ($k1$ and $k2$) were defined as important muscle fatigue indexes. The recovery of normalized average force after 1 min rest (1MinRec) represented the fatigue recovery ability of the elbow flexor muscle.
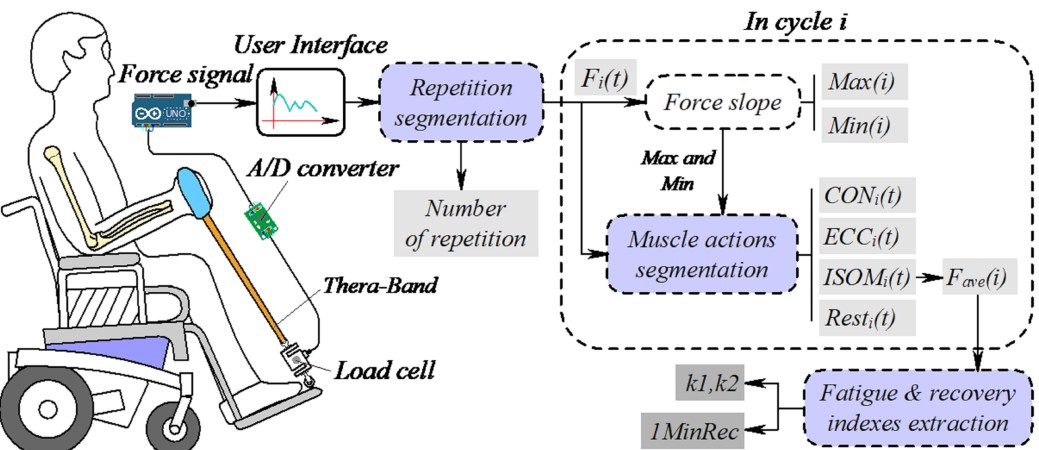

**Figure 1 Concepts and big picture of the system.** $F_i(t)$ is the resistance in repetition $i$. Max($i$) and Min($i$) are the maximum and minimum force slope values. CON, ECC and ISOM are concentric, eccentric and isometric contraction, respectively. Rest is the force data in the rest phase, $F_{ave}(i)$ is the average force during ISOM, used to represent the fatigue and recovery profiles of elbow flexor during dynamic contraction and $k1$ and $k2$ are the decreasing slopes of $F_{ave}(i)$ before and after rest. 1MinRec is the recovery of $F_{ave}(i)$ after 1 min rest. Drawing source credit: Taojin Xu.

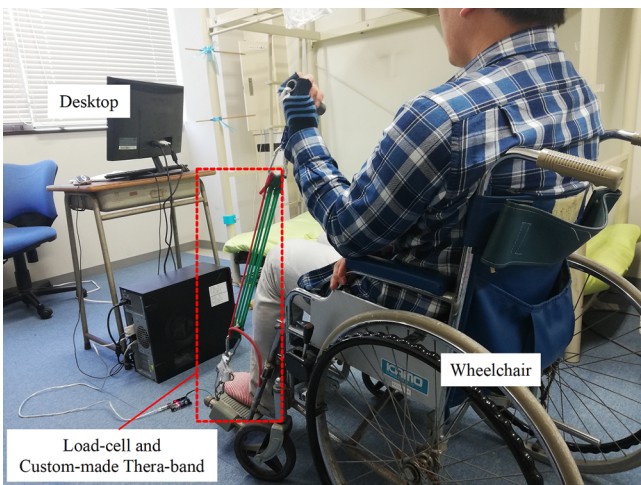

**Figure 2 The experiment set-up.** Photo source credit: Taojin Xu.

## Measuring system

Figure 2 shows the experiment set-up. A standard wheelchair is utilized to help the subject sit in a stable position and to help him/her lean the left or right arm against the armrest of the wheelchair. The subject flexes his/her forearm in the sagittal plane against the resistance produced by the custom-made resistance band. Figure 3 illustrates details of the sensor, band and Arduino board. In resistance training, the subject pulls the handle which is connected to the band. The band is attached to the load cell, which is anchored to the footplate of the wheelchair by a lifting hook. As mentioned above, a load cell

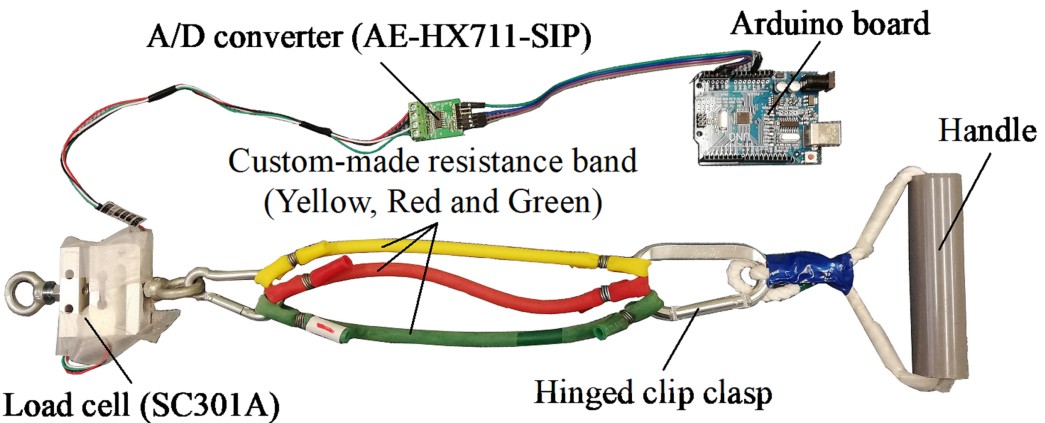

**Figure 3 Details of the sensor, custom-made Thera-Band and Arduino board.** Photo source credit: Taojin Xu.

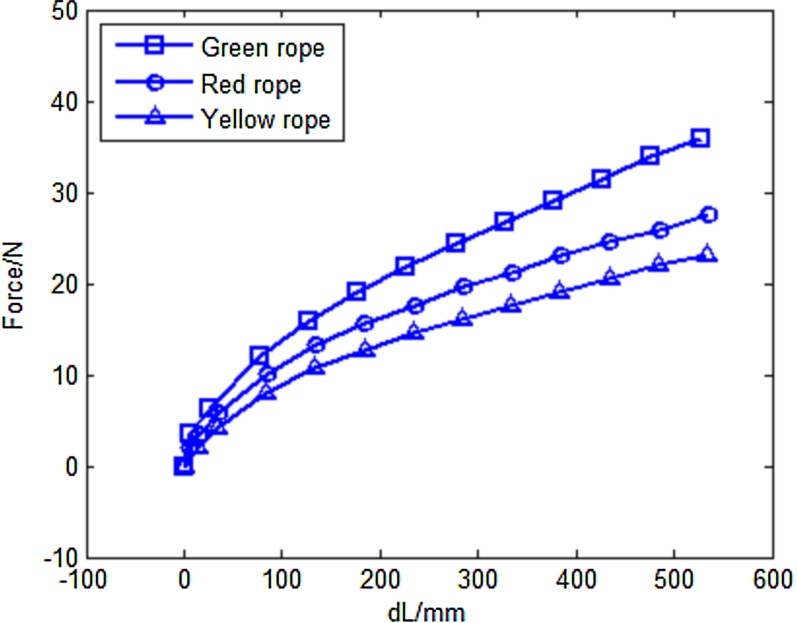

**Figure 4 Load versus extension curves of the different custom-made Thera-Bands.**

(SC301A, 100 kg) was used to measure the time-varying resistance. Analog signals were converted to digital signals through an A/D converter (AE-HX711-SIP, 30 Hz). The load cell was calibrated and the scale factor was calculated before the experiments using calibration weights. A personal computer (PC) was used to acquire, display, save and process the force data through a serial COM port. Since the training load was different among different trainees, three different types of band (Green, Red and Yellow) were constructed and connected to the handle using a carabiner clip. As shown in Fig. 4, each band was tested under uniaxial tensile extension and the force data were recorded to obtain

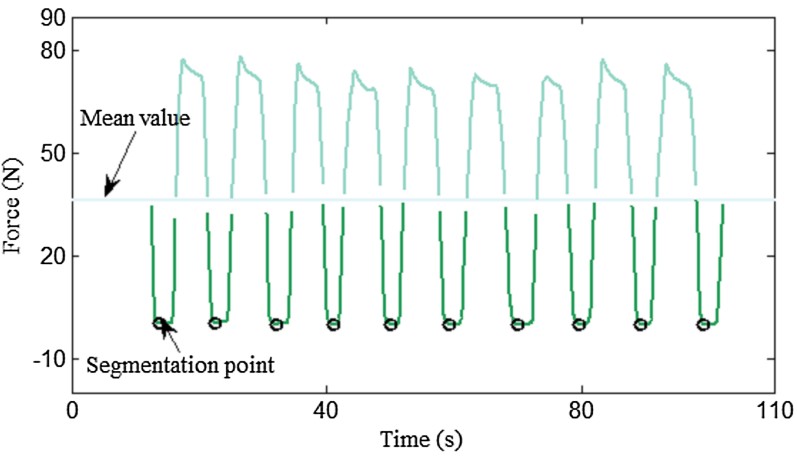

**Figure 5 Segmentation points locating logic.** The minimum value of the trough was chosen as the segmentation point.               

the load vs. extension curve. Based on the curves, it was observed that the green band had the highest resistance, while the yellow band had the lowest. In our study, different combinations of bands were used to set different training loads for the trainees according to their muscle strength.

## Force data segmentation

The proposed algorithm incorporates repetition segmentation and muscle action segmentation. The repetition segmentation is a pre-processing stage, where the force data are divided into individual segments according to the training cycle. Figure 5 illustrates a typical plot of resistance data during the biceps curl exercise. As shown by the curve, the force signal exhibits strong periodicity with distinct peaks and troughs. A simple and efficient peak detection method (*Chen et al., 2015*) was used to find the minimum value of the trough, which was taken as the segmentation point. In the segmentation process, the mean resistance value was used as the threshold value to divide the force signal into peaks and troughs. An example of segmentation results is illustrated in Fig. 6.

Subsequently, a muscle action segmentation algorithm was proposed to divide each biceps curl cycle into four time-windows. The biceps curl exercise is an easy-to-recognize movement, which incorporates all the CON, ISOM and ECC muscle actions. However, in general there is no clear cut-off point between CON, ISOM, ECC and rest phases, since training is a complex process that is, affected by many subjective factors, such as personal ability, willingness and fatigue. Figure 7 presents a visual representation of how the segmentation algorithm operates. The core idea of the segmentation algorithm is based on the difference in force derivatives at the different contraction phases. Raw force data containing one flexion-extension repetition were obtained using the repetition segmentation algorithm proposed in the previous paragraph. Firstly, the first order derivative of force vs. time was numerically calculated. From the derivative curve, it was perceived that the maximum and minimum values were the most discriminating features.

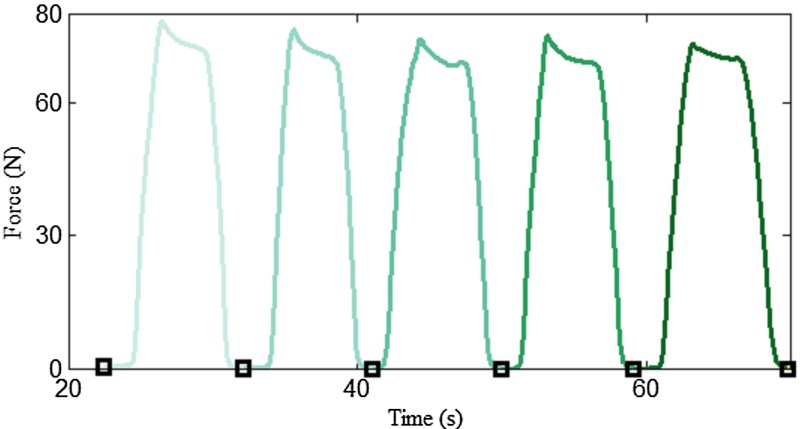

**Figure 6 Example of a segmentation result.** Different training cycles are illustrated in different colors.

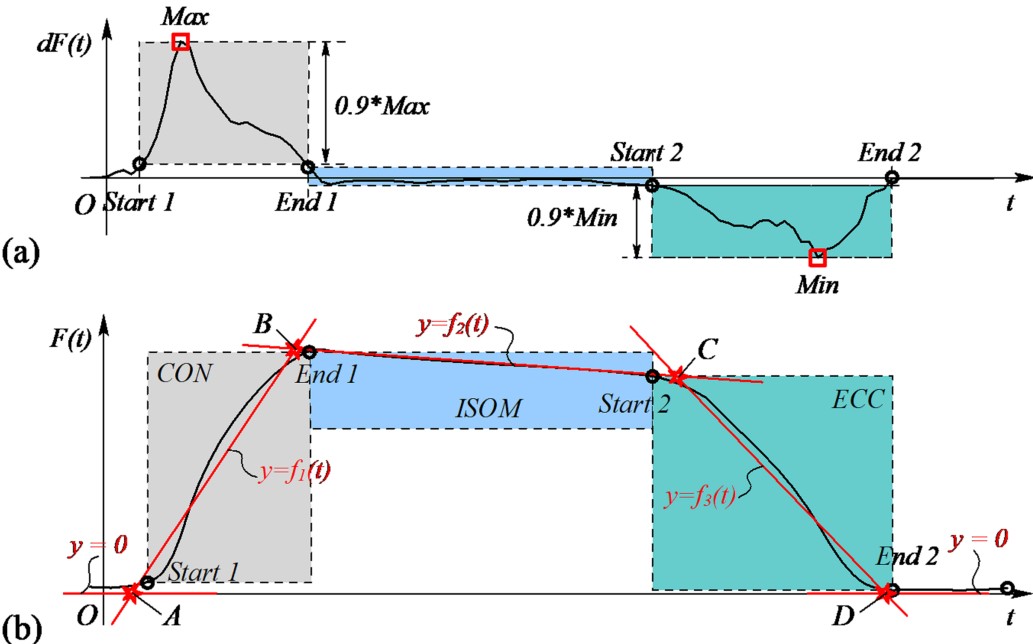

**Figure 7 Schematic diagram of the segmentation algorithm.** (A) The maximum and minimum values of the first order derivative of force vs time were used as thresholds to divide the data into three parts. (B) Trend line intersections were used to segment the arm biceps curl exercise into four contraction phases (CON, ISOM, ECC and rest).

During the CON contraction, the force-derivative increases rapidly at the beginning, reaches a maximum value and then rapidly decreases to zero. In ISOM contraction, the subject keeps his/her forearm in quasi-static state for a few seconds and the force-derivative does not change much. During the ECC contraction, the subject releases his/her forearm and the force-derivative reaches a minimum value. Consequently, the

maximum and minimum values of the force-derivative were considered as important thresholds to segment the force data. The segmentation model can be expressed as follow:

$$F(t) \in \begin{cases} CON, & (0.1 \, ^* \, \text{Max} \leq dF(t) \leq \text{max}) \\ ISOM, & (0.1 \, ^* \, \text{Min} < dF(t) < 0.1 \, ^* \, \text{max}) \\ ECC, & (\text{Min} \leq dF(t) \leq 0.1 \, ^* \, \text{Min}) \end{cases} \tag{1}$$

During the CON and ECC contraction, data with a derivative greater or less than 10% of the maximum and minimum values were chosen and the trend line was calculated to approximate the change in force during these contraction phases. The trend line is an approximate alternative of the data in the contraction phases. On the one hand, this approximation eliminates some accidental situations in the data, such as fluctuations or slacks. On the other hand, when the forearm flexes or extends, the force tendency is preserved as much as possible. The trend line ($y = f_i(t)$, $i = 1–3$) was calculated using linear regression. The trend line intersections (A–D) and the line of constant function ($y = 0$) divided the arm biceps curl exercise into the CON, ECC and ISOM contraction phases and the rest period.

## Extraction of fatigue and recovery indexes

In general, exercise is accompanied by muscle fatigue. When fatigue occurs, the threshold to trigger action potentials in a motor unit increases, that is, the motor unit's tendency to fire decreases (*Liu, Brown & Yue, 2002*; *Enoka & Fuglevand, 2001*). Since the discharge rate changes, the number of recruited motor units is reduced, resulting in the reduction of muscle force. Here, muscle fatigue was defined as the relative decline in average force ($F_{ave}(i)$) during ISOM contraction and the decreasing slopes ($k1$ and $k2$) of $F_{ave}(i)$ were used as index of fatigability. The recovery of $F_{ave}(i)$ after 1-min rest (1MinRec) was used to quantify subject's ability to recover from fatigue.

Figure 8 presents how the decreasing slopes and the recovery of the average force after rest were calculated. The average force during ISOM contraction is expressed as follow:

$$F_{ave}(i) = \frac{B_i(y) + C_i(y)}{2} \tag{2}$$

where $B_i(y)$ and $C_i(y)$ represent the $y$ value of the starting and ending point of ISOM, respectively. The regression slopes $k1$ and $k2$ were calculated using a regression function. The calculation of 1MinRec is expressed as follow:

$$1MinRec = \frac{P2(y) - P1(y)}{F0} \times 100\% \tag{3}$$

where $P2(y)$ and $P1(y)$ represent the $y$ value of points $P1$ and $P2$, respectively, and $F0$ represents the initial force.

## Experimental procedures

Figure 9 gives a visual representation of the procedure followed in the fatigue and recovery protocol. The protocol has been carefully designed to incorporate four parts: one-repetition maximum (1RM) test, 5 min ISC, 1 min rest, and 2 min ISC. The 1RM test

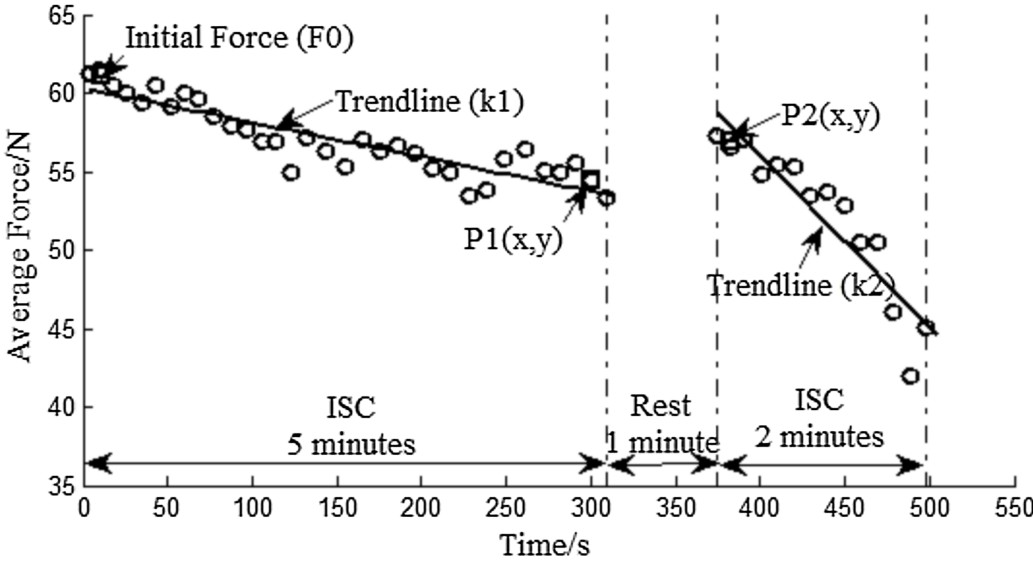

**Figure 8 Average force distribution and regression trend line of a male subject.** Initial force ($F0$) and $P1(x, y)$ are the mean value of the first and last three data during the 5 min ISC. $P2(x, y)$ is the mean value of the first three data during the 2 min ISC.

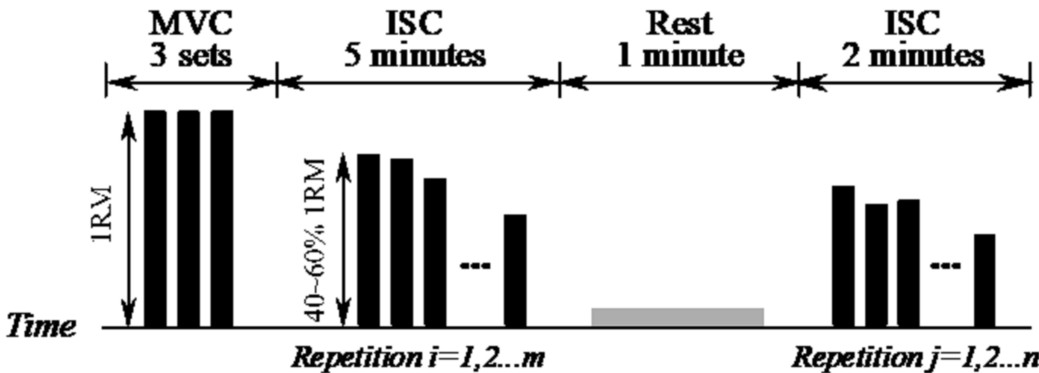

**Figure 9 Schematic of the experimental procedure.** 1RM is the one-repetition maximum test, MVC is the maximum voluntary contraction and ISC is the intermittent submaximal contraction. Each black bar represents 1 repetition, which incorporates the CON, ISOM and ECC contractions.

is performed to assess subject's flexor muscle strength and the obtained value is the basis for selecting the combination of elastic bands. During the 1RM test, the number of bands was empirically increased to reach the maximum load the subject can pull while flexing his/her forearm throughout the full range of motion (elbow joint angle more than 150°). Three sets of contractions were performed and the average force was calculated as the final 1RM. During the 5 min ISC, the number of elastic bands was chosen to ensure that the initial average force was 40–60% that of 1RM. ISC is an intermittent and dynamic voluntary contraction process with many repetitions. During each repetition, the subject flexes his/her forearm to a maximum angle (CON contraction), maintains the angle for about 3 s (ISOM contraction), and then returns to a relaxed state

(ECC contraction). A short rest was allowed between each repetition and the rest time was usually no more than 4 s. During the 1 min rest, the subject was encouraged to stand up from the wheelchair and swing his/her arm to recover from the fatigued state. The 1 min rest and 2 min ISC were performed in succession after the 5 min ISC. The subjects performed the exercises with both hands and were given ample rest between experiments. All subjects were right-handed. A training session was conducted in order to get the subjects familiar with the experimental procedure before the formal experiment. In addition, the subjects performed the biceps curl exercise against resistance band during the 5 min and 2 min ISC. The biceps curl movement consisted of the motion of elbow joint flexing and extending while the wrist joint was fixed in a neutral position (no flexion or extension). Flexion represented the movement of elbow flexion and extension represented the movement of elbow extension.

At this stage, we aimed to see if the actual monitoring system was able to detect the contraction phases and the ability of the subjects to use the prescribed TUT in the ISOM phase (3 s). Therefore, in this study, the experiment instructor was required to accompany the subject to complete the entire experiment. During the experiment performed by each subject, the instructor was required to start the measurement system, click the "Record" button and remind the subject about the start and end time of the entire experiment, as well as the 1 min rest period. Subjects were told to keep the TUT in the ISOM phase for 3 s and a metronome was used to guide on the TUT in ISOM phase. The metronome provided guidance for the subject, not the instructor.

## Statistical analyses

Statistical analysis was carried out in Excel (Microsoft Office 2010, Microsoft, Redmond, Washington D.C., USA) and with the aid of MATLAB (R2012b). Descriptive statistics were performed to subjects' demographic data. The AVERAGE and STDEV.S functions were used to calculate the mean and standard deviation of subjects' personal information, respectively. The contraction-phase specific TUT was calculated using MATLAB. Student's test was performed to compare the difference in decreasing slope and 1MinRec between left and right hand. Significance test for the alpha value was set at 0.05.

## Participants

Some formalities were required to obtain an ethical license to allow experiments with the elderly population. Moreover, unlike healthy individuals, the equipment used by older individuals needs to be safer and more reliable. Therefore, even the case of this article was particularly made for an older demographic, healthy subjects were recruited to show the performance of our segmentation algorithm. Six healthy subjects (3 males and 3 females, age $27.3 \pm 4.2$ years, weight $60.3 \pm 11.7$ kg, height $171 \pm 9.7$ cm) volunteered to participate in this pilot experiment. Participant recruitment and experiments were conducted in the affiliate of the first author, Micro Mechatronics Laboratory, Faculty of Civil Aviation and Aeroautics, Kunming University of Science and Technology, Kunming, China. All subjects were familiarized with the research procedures and gave their
| | Subject no. | Age (year) | Weight (kg) | Height (cm) |
|---|---|---|---|---|
| Female | S01 | 34 | 50 | 170 |
| | S02 | 24 | 57 | 159 |
| | S03 | 27 | 48 | 160 |
| Male | S04 | 28 | 65 | 178 |
| | S05 | 22 | 62 | 182 |
| | S06 | 29 | 80 | 177 |

**Table 1 Demographic data of the study participants.**

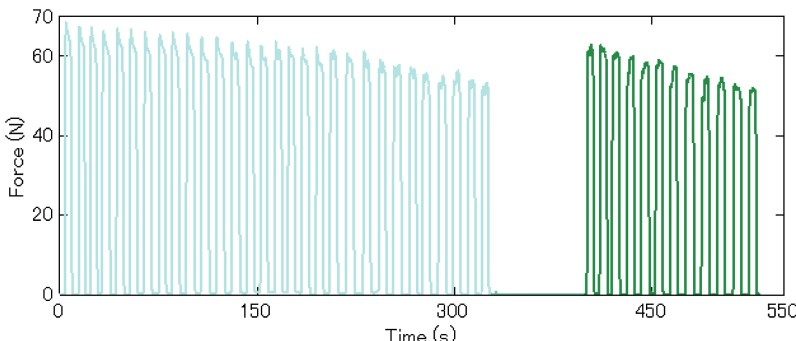

**Figure 10 Representative data of a male subject.** The light blue curve represents data during the 5 min ISC and the green curve represents data during the 2 min ISC.

informed consent. None of the subjects reported known neurological, musculoskeletal, or orthopedic disorders. Demographic data of the study participants are presented in Table 1.

# RESULTS

Figure 10 shows a typical force-time curve of a male subject. As dynamic contraction progressed, the maximum force the subject could reach reduced gradually and was restored to a certain extent after the 1 min break. When participants became exhausted, there were some fluctuations in force during ISOM contraction. Other details about the experiment, including the initial force, are presented in Table 2. As described in the previous section, the combination of resistance bands was chosen based on a subject's muscle strength. Furthermore, the repetition numbers before rest (m) and after rest (n) were calculated and are presented in Table 2.

## Contraction-phase specific TUT

The proposed segmentation algorithm was successfully applied with the aid of MATLAB and after all participants had completed the experiment, TUT was calculated. The calculated contraction-phase specific TUT could be applied to investigate whether participants followed the prescribed home-based program and to evaluate performance of the segmentation algorithm. Table 3 reports the proportion of contraction-phase specific TUT and the total TUT of all subjects. The total TUT is the sum of TUT during CON, ISOM and ECC contractions. The proportion of TUT varied greatly between subjects.

**Table 2 The 1RM initial force combination of Thera-Bands and repetitions in the experiment.** The 2R means subject used two Red Thera-bands in exercise.

| | Subject no. | Hand | 1RM ($N$) | Initial $F_{ave}$ ($N$) | Combination of Thera-bands | Repetitions | |
|---|---|---|---|---|---|---|---|
| | | | | | | $m$ | $n$ |
| Female | S01 | Left | 87.8 | 45.1 | 2 R | 40 | 17 |
| | | Right | 91.6 | 58.4 | 1 G + 1 R | 37 | 20 |
| | S02 | Left | 67.1 | 42.0 | 1 R +1 Y | 39 | 18 |
| | | Right | 59.5 | 44.7 | 1 R +1 Y | 41 | 16 |
| | S03 | Left | 108.6 | 43.0 | 1 G + 1 R | 49 | 19 |
| | | Right | 111.2 | 47.2 | 1 G + 1 R | 45 | 23 |
| Male | S04 | Left | 152.5 | 82.5 | 3 R + 1 Y | 41 | 16 |
| | | Right | 163.2 | 91.3 | 3 R + 1 Y | 40 | 17 |
| | S05 | Left | 132.2 | 65.3 | 2 R + 1 Y | 30 | 12 |
| | | Right | 140.8 | 73.4 | 3 R | 32 | 10 |
| | S06 | Left | 193.6 | 112.8 | 4 G +1 R | 37 | 17 |
| | | Right | 189.6 | 120.6 | 4 G +1 R | 35 | 19 |

**Table 3 Proportion of contraction-phase specific TUT and the total TUT before and after rest.** Total TUT are the sum of TUT during CON, ISOM and ECC contraction. L denotes left hand and R denotes right hand.

| Subject no. | | Before rest | | | | | After rest | | | | |
|---|---|---|---|---|---|---|---|---|---|---|---|
| | | CON* | ISOM* | ECC* | Rest* | Total** | CON* | ISOM* | ECC* | Rest* | Total** |
| S01 | L | 18.1 | 33.2 | 26.7 | 22.0 | 249.6 | 20.1 | 32.1 | 23.3 | 24.4 | 102.1 |
| | R | 18.3 | 33.3 | 22.8 | 25.6 | 229.4 | 17.9 | 34.3 | 26.0 | 21.7 | 105.5 |
| S02 | L | 11.5 | 49.9 | 8.0 | 30.7 | 216.7 | 10 | 52 | 7.6 | 30.4 | 93.9 |
| | R | 11.3 | 44.3 | 8.3 | 36.2 | 203 | 10.4 | 41.1 | 8.5 | 40.0 | 81.5 |
| S03 | L | 13.4 | 67.7 | 10.3 | 8.6 | 297.3 | 14.2 | 67.2 | 9.6 | 9.0 | 119.1 |
| | R | 14.1 | 68.5 | 11.3 | 6.1 | 295.6 | 13.3 | 67.1 | 12.3 | 7.4 | 120.7 |
| S04 | L | 10.3 | 50.7 | 10.7 | 28.3 | 223.2 | 9.3 | 49.5 | 9.2 | 31.9 | 85.5 |
| | R | 11.6 | 50.3 | 12.4 | 25.7 | 228.8 | 10.6 | 48.4 | 10.5 | 30.4 | 87.1 |
| S05 | L | 10.8 | 37.9 | 8.4 | 42.9 | 179 | 12.3 | 37.6 | 9.4 | 40.8 | 78.8 |
| | R | 10.9 | 35.4 | 9.6 | 44.1 | 171.5 | 11.5 | 38.9 | 10.8 | 38.8 | 69.2 |
| S06 | L | 8.4 | 45.7 | 11.6 | 34.4 | 203 | 9.4 | 44.6 | 9.6 | 36.4 | 92 |
| | R | 10.1 | 44.7 | 7.6 | 37.7 | 191.7 | 11.2 | 43.7 | 8.5 | 36.6 | 89.8 |

**Notes:**
* Data is expressed in percentages.
** Seconds.

The subjects received very different contraction-phase specific and total TUT, even though the total training time was the same. Differences in TUT make it difficult for clinicians and researchers to interpret the training results and state if the subject receives the prescribed training volume (*Faber et al., 2015*). In addition, even if a metronome was used to provide guidance on the TUT, the average TUT (3.75 ± 0.62 s) during ISOM was longer than 3 s, meaning that most of the subjects did not perform the exercise as prescribed

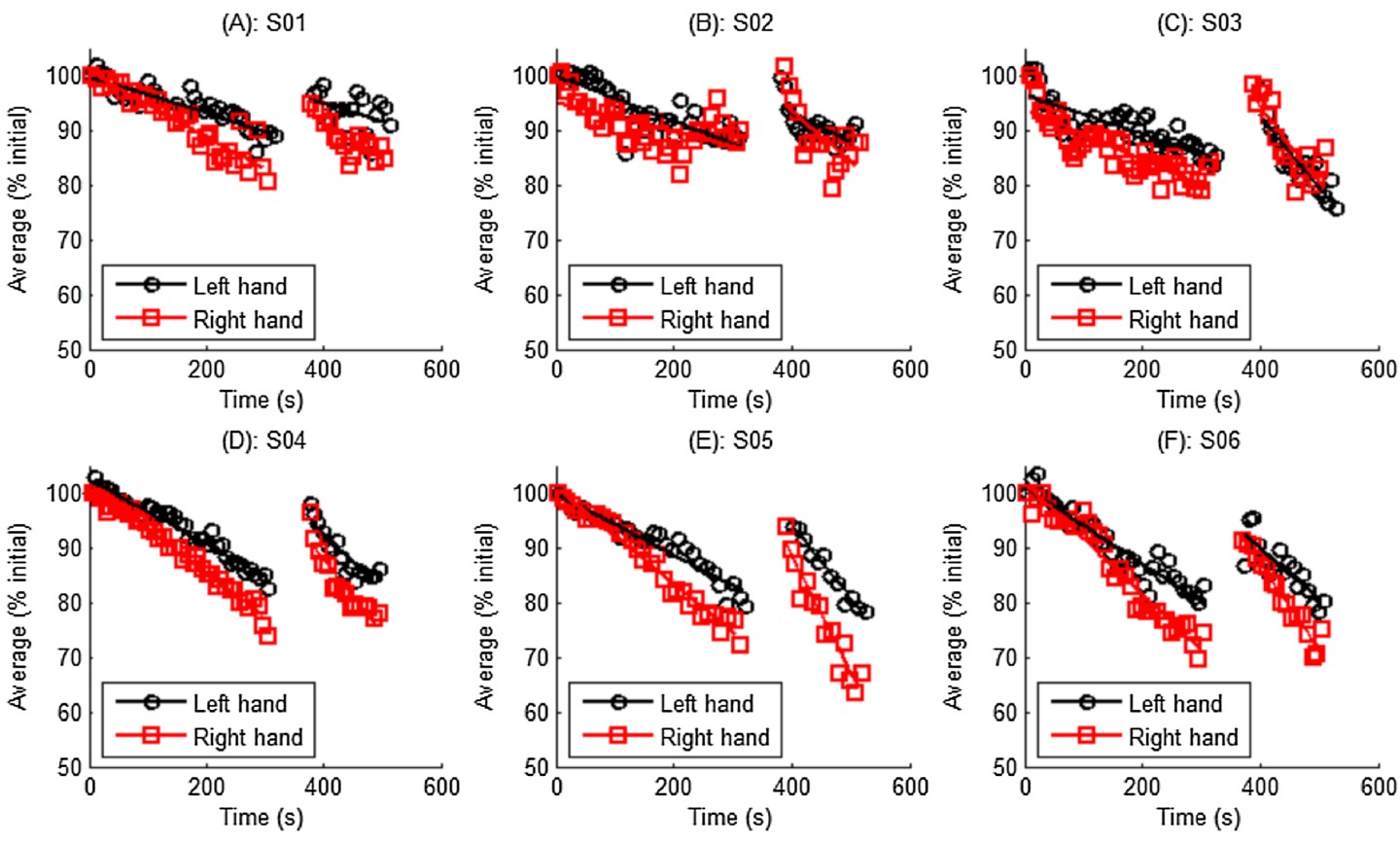

**Figure 11 Normalized average force distribution and trend lines for all participants.** (A) Data of subject S01. (B) Data of subject S02. (C) Data of subject S03. (D) Data of subject S04. (E) Data of subject S05. (F) Data of subject S06.

(*Faber et al., 2015*). The differences in TUT emphasize the importance and necessity of monitoring the exercise dose during home-based resistance training.

## Elbow flexor fatigue and recovery profiles

The acute fatigue response could be investigated to explain how fatigue experience affected the compliance of patients as it is believed that fatigue experience may decrease a patients' motivation to participate in a training program (*Andrew & Parker, 1979*; *Hughes et al., 1984*). Here, we present the calculated elbow flexor fatigue and recovery profiles during the training to prove the capability of the system in monitoring the acute fatigue response. The average force during ISOM contraction was calculated using the aforementioned algorithm. The force data distributions and the resulted trend lines for all participants are shown in Fig. 11. For the purpose of comparison, all force data were normalized to the initial value (*Callahan, Umberger & Kent, 2016*; *Sundberg et al., 2018*; *Thompson et al., 2014*; *Lanza et al., 2006*). The decreasing slopes of the average force before and after rest in female and male subjects were analyzed and the statistical result is given in Table 4. Table 5 reports the statistics of average force recovery in left and right hand after 1 min rest.

It shown in Fig. 11, and stated in many references (*Callahan, Umberger & Kent, 2016*; *Sanchez-Medina & González-Badillo, 2011*), all definitions of fatigue necessitate a decline

**Table 4 Statistics of the decreasing slope of average force before and after rest.** The smaller the slope, the faster the average force drops. Student test was performed to examine the decreasing slope difference between left and right hands.

| | Decreasing slope of average force | |
| --- | --- | --- |
| | **Before rest ($k1$)** | **After rest ($k2$)** |
| Female | $-0.0407 \pm 0.0115^*$ | $-0.0860 \pm 0.0416^*$ |
| Male | $-0.0758 \pm 0.0176^*$ | $-0.140 \pm 0.0425^*$ |
| Overall | $-0.0582 \pm 0.0231^*$ | $-0.113 \pm 0.0492^*$ |
| *T*-test | $p = 0.2^{**}$ | $p = 0.14^{**}$ |

Notes:
  [*] Data is expressed as mean ± standard deviation.
  [**] $p > 0.05$.

**Table 5 Statistics of 1MinRec in left and right hand.**

| | **1MinRec (% Initial)** |
| --- | --- |
| Left hand | $9.968 \pm 1.889$ |
| Right hand | $13.738 \pm 3.149$ |
| Overall | $11.853 \pm 3.163$ |
| *T*-test | $p = 0.03^*$ |

Notes:
  [*] $p < 0.05$.
  Data are expressed as mean ± standard deviation; Student test was performed to examine the 1MinRec difference between left and right hands.

in force or power, the average force during ISOM contraction decreased continuously. The Student's *t* test revealed that no significant differences were observed in the decreasing slope between left and right hands ($p = 0.2$ and $p = 0.14$, respectively, Table 4). The average force restored to a certain extent after the 1 min rest, however it decreased much faster than before the 1 min rest. This was probably because the 1 min rest does not prevent the development of central fatigue (*Bilodeau, 2006*).

## DISCUSSION

In the previous sections, the measurement system was described and a segmentation algorithm that automatically segments the biceps curl cycle into small parts based on muscle actions was proposed. A pilot experiment with six subjects was conducted and the repetition number and contraction-phase specific TUT were calculated and compared. Some fatigue and recovery indexes were proposed in order to show fatigue-recovery profiles of elbow flexor for all subjects. Furthermore, in the following sections, the performance of the segmentation algorithm was evaluated and the application prospect of the instrument in a practical setting was discussed.

### Performance of the segmentation algorithm

To the best of our knowledge, this is the first study that uses an automated method to rate the contraction-phase specific TUT. Compared to other manual methods (*Faber et al., 2015*; *Rathleff Skovdal, Thorborg & Bandholm, 2013*), this method exhibited high

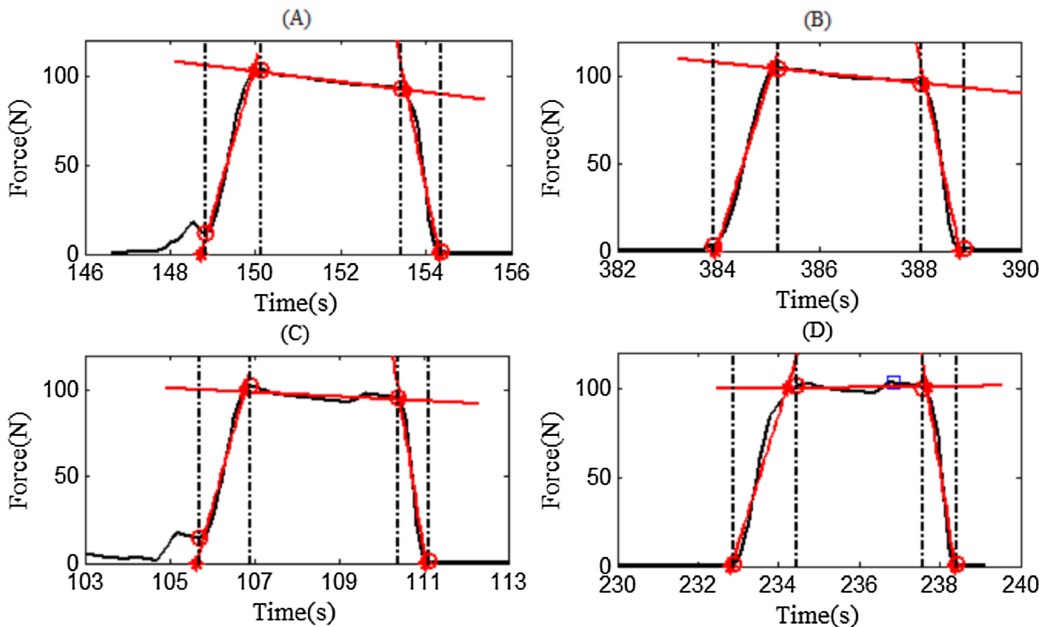

**Figure 12 Four representative examples of segmentation results.** Red asterisks denote intersections and red circles denote segmentation points. The trendlines are denoted by the red lines. (A) Representative example of subject S01. (B) Representative example of subject S01. (C) Representative example of subject S01. (D) Representative example of subject S01.

efficiency. Using a dual-core 3.5 GHz Intel processor laptop, the rating time for one trial takes no more than 5 s. Some typical segmentation results are presented in Fig. 12. As demonstrated, the proposed algorithm can correctly segment the force data into the four contraction phases (CON, ISOM, ECC and Rest), even when there are some fluctuations or slacks between two contraction phases. In some cases, when the trainee bent his/her arm at the beginning of CON, some failed startups occurred. During those failed startups, the resistance was relatively small and these data did not produce effective TUT. In such cases, the algorithm removed these invalid data and used the trend line to determine the most likely starting point. In other cases, the boundary between two contraction phases was clearly marked by the trend line intersection. The segmentation results were visually checked and it was found that the algorithm successfully segmented all the data.

To further verify the correctness of the algorithm, two assessors, who were unfamiliar with the experiment, were recruited to manually pick the visually observed cut-off point between two contraction phases and the calculation results of TUT from the two assessors were used as gold standard for assessing the TUT results from our segmentation algorithm. The TUT comparison is demonstrated in Fig. 13 and shows good TUT agreement and contraction segment agreement between the proposed algorithm and the assessor. The total TUT was exactly the same, however the algorithm's TUT was a little longer than that of the assessors during CON and ECC, while during ISOM, the algorithm's TUT was a little shorter than that of the assessors. In addition, the comparison
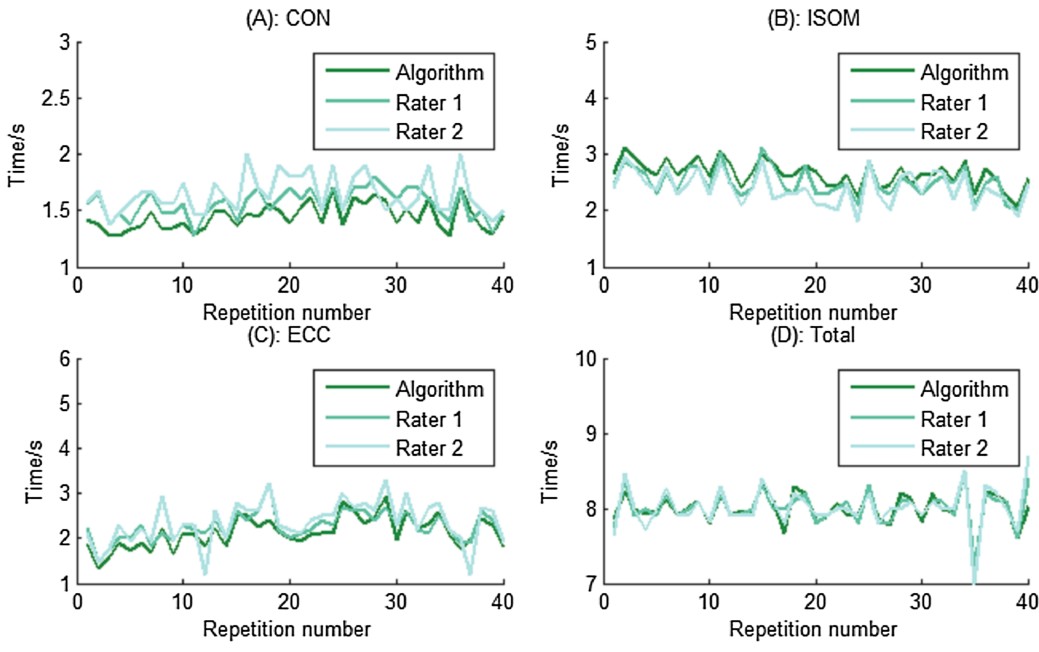

**Figure 13 TUT and contraction segments comparisons for one participant at different phases.**
The results show good TUT agreement and contraction segment agreement between the proposed algorithm and the assessor. (A) TUT during CON phase. (B) TUT during ISOM phase. (C) TUT during ECC phase. (D) The total TUT.

results showed that there were differences in the rating results between assessors, thereby indicating that the rating of the data was subjectively influenced by the assessor.
The proposed segmentation algorithm was implemented programmatically, avoiding the subjectivity of the assessors. Therefore, it is an automated and objective method for quantifying the exercise dosage during home-based elbow flexor resistance training.

## Application prospect of the instrument in a practical setting

The hypothetical application scenario of this instrument is to help people who were confined to a wheelchair or bed to perform elbow flexor resistance training. Compared with healthy subjects, those people have deficits in their physiological system and are more sensitive to the training load and volume. Consequently, a custom-made resistance band shown in Fig. 3 was designed and made to monitor the training dose and acute fatigue responses. In this study, we only showed the example of using biceps curl exercise with resistance bands to regain elbow flexor muscle strength in people who were confined to a wheelchair or bed. However, this system can be applied easily to other more functional movements, including forearm extension or elbow extensor resistance training exercise. When a trainee holds the resistance band and performs training exercise, an emerging problem is how to fix the other end. By changing the way how the other end is fixed, this method can be applied to many other movements. Moreover, as presented in Fig. 3, the measuring system is easy to use and cheap, indicating its broad application prospects. Presently, only two resistance bands were made, one is applied for testing at an

elderly rehabilitation center. Additional bands will be prepared with the assistance of sporting goods companies to meet the safety and reliability requirements of rehabilitation centers for elderly people. In addition, the segmentation algorithm was implemented in MATLAB. A website version of the segmentation algorithm will help more people to freely and conveniently use this algorithm. Moreover, we are currently designing a novel measuring system, which allows for automatically uploading the resistance force to the website through a single-board computer. Through the novel measuring system and website interface, therapists will be able to remotely monitor the training dose of the home-base trainee and more conveniently obtain experimental data. In conclusion, additional studies will be needed before our approach can be applied to a wider range of practical applications.

### Limitations and future work

A limitation of this pilot study was its small sample size in the experiment. Only six healthy subjects volunteered to participate in the experiment, and the small sample size may have weakened the application of some conclusions to a larger population. With additional time and funding, future studies with larger sample size, including participants from a wider population, will address this limitation. In addition, the final goal of this project was to establish an automatic monitoring system for home-based resistance training. At this stage, we aimed to see if the actual monitoring system was able to detect the contraction phases and detect the ability of the subjects to use the prescribed TUT in the ISOM phase (3 s). We did not investigate how the fatigue affects compliance of patients. However, since the fatigue experience will affect patients' compliance to training programs, how and to what extend the fatigue experience affect patients' compliance will be investigated in our future work.

## CONCLUSIONS

In summary, this study presented an automated method for segmenting resistance data into contraction phase-specific segments during elbow flexor resistance training, for establishing an automatic monitoring system for home-based resistance training. The principle of the algorithm was described in detail and experiments were performed to evaluate the performance of the method. The good agreement in TUT measurements between the proposed algorithm and the assessors indicated that the proposed algorithm can correctly segment the contraction into contraction phase-specific parts, thus providing clinicians and researchers with an automated and objective method for quantifying home-based elbow flexor resistance training. The objective and automated nature of the segmentation algorithm has the advantage of eliminating the subjective influence of the assessor and improving the rating efficiency. In addition, some fatigue and recovery indexes were proposed in order to show fatigue-recovery profiles of elbow flexor.

## ACKNOWLEDGEMENTS

The authors are very grateful to Msrh Izza and Lurui Wang for rating the experimental data.

### Funding

This research is supported by National Natural Science Foundation of China (No. 51865021) and Scientific Research Foundation (No. cstc2017rgzn-zdyf0054) from Chongqing Municipal People's Government. There was no additional external funding received for this study. The funders had no role in study design, data collection and analysis, decision to publish, or preparation of the manuscript.

### Grant Disclosures

The following grant information was disclosed by the authors:
National Natural Science Foundation of China: 51865021.
Scientific Research Foundation: cstc2017rgzn-zdyf0054.

### Competing Interests

The authors declare that they have no competing interests.

### Author Contributions

- Jingjing Yang conceived and designed the experiments, performed the experiments, analyzed the data, prepared figures and/or tables, authored or reviewed drafts of the paper, and approved the final draft.
- Hongbin Xu conceived and designed the experiments, authored or reviewed drafts of the paper, and approved the final draft.
- Juke Liang conceived and designed the experiments, performed the experiments, analyzed the data, prepared figures and/or tables, and approved the final draft.
- Jongyeob Jeong conceived and designed the experiments, analyzed the data, prepared figures and/or tables, authored or reviewed drafts of the paper, and approved the final draft.
- Taojin Xu analyzed the data, prepared figures and/or tables, authored or reviewed drafts of the paper, and approved the final draft.

### Human Ethics

The following information was supplied relating to ethical approvals (i.e., approving body and any reference numbers):

The Medical Ethics Committee in Kunming University of Science and Technology approved this research (approval number 2019YXY06).

### Data Availability

The raw data is available in the Supplemental Files.

The code (which is not a product of the research) cannot be provided for intellectual property reasons, as it is owned by a third party (MicroMechatronics Lab. of Engineering Faculty in Yamaguchi University). Interested readers may obtain the code by contacting Lurui Wang at b001wc@yamaguchi-u.ac.jp.

## Supplemental Information

Supplemental information for this article can be found online at http://dx.doi.org/10.7717/peerj.8689#supplemental-information.

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
