# Peer review of "Monitoring the training dose and acute fatigue response during elbow flexor resistance training using a custom-made resistance band"

_PeerJ, doi:10.7717/peerj.8689_

## Round 0.1 · original submission · Major Revisions

I have received two reviewers of your manuscripts. Both reviewers agree that your work conveys interesting data, although they report a number of flaws that require your full attention. An important point is that your main goal is not clearly expressed. Reviewer #1 expressed concerns in relation to the lack of a prescribed program to be used by a therapist. Reviewer #2 pointed out a major problem in that you speculate too much about factors that you have not controlled and for which you don’t have adequate methods, like fatigue and strength. He strongly suggests removing this from the study and concentrates on the use of the procedure you present, and I concur. Please, stick to discuss your own data.

Please review your work carefully in light of all the suggestions of our reviewers.

·

Basic reporting

In general, the manuscript is well written although it could do with one more edit as I found a few problems with the writing and clarity of wording. e.g. line 376 "bended" instead of "bent"; line 378 should be "did not" rather than"do not". There are a few others that I am sure the authors will identify with another read of the manuscript. "Theraband" is a trade name. Did you actually use "therabands" or some other tubing.

I thought the study was well conducted and showed some good internal validity, especially in determining the training loads using "Thera-bands". This was carefully controlled in the experimental setting. However, when most home-based programs are prescribed the patient is given a "Theraband" usually of a specific colour and expected to do the exercises at home. They are able to set their own tension based on how much band they work with. So your study is much more precise in setting the work load or resistance. Would you expect this approach be applied in a practical setting? You may wish to comment on this aspect of the study and its practical application. However, my main concern is the lack of clarity of the actual study as the main purpose seems to get lost in a number of peripheral questions. It seems to me the main purpose is to see if "patients" can follow a prescribed home-based program on strength training or rehabilitation. In order to do this you have devised an instrument that potentially can be used by a therapist to check if the patient has followed a prescribed program. The study shows that the instrumentation you devised is able to show the different phases of muscle contraction such as concentric, eccentric, isometric and the rest periods using the algorithm you developed. What is less clear to me is the question of whether the subjects followed the prescribed program. In fact, I could not find if an actual program was prescribed. There is some reference a metronome was used but not sure if this was in the familiarization session or the subjects were allowed to use it when theoretically doing the exercises by themselves. Normally when prescribing a program for strength, the resistance is set and a number of repetitions related to the % of the 1RM along with a tempo for the various segments of the contraction. In this study the program seems to consist of intermittent sub-maximal contractions (ISC) repetitions at a set load for 5 min, then a 1 min rest, followed by another 2 min. There appears little direction in regard to the tempo or repetitions. Were the subjects directed to resist the lengthening or eccentric phase? It seems to me a more ecologically valid study would be specific in prescribing a program and seeing if patients were able to follow it and this would be the crux of the question. Or have I missed something? Please clarify if this is the case.

I did have a lot of trouble relating the text to the figures. It would have helped to have clear figure captions as well as the number of the figure shown on the actual figure. I am not sure if this is a journal requirement but from a reviewer perspective it would make it much easier to find the appropriate figure when reading the text. I ended up having to count the figures as well as scroll down a number of times. Do you need all the figures and tables. I think you could reduce the number if you narrowed the focus of the study and concentrated on the main purposes.

Experimental design

You provide a good rationale for the study and the need to be able to assess the compliance of patients in following prescribed programs to regain or increase their strength. The case is particularly made for an older demographic so I was surprised when you used subjects between 22 and 34 years of age. If your interest is in an older cohort why did you not use older subjects? Older subjects will respond very differently to a younger population in regard to exercise and force application.

I also feel a lot of the discussion related to age, gender, and weight is limited based on the number of subjects and does not add to the main purpose of the study. I appreciate the fatigue discussion is interesting and from a training perspective is important but not sure if it is relevant to the main purpose of the study unless you provide a stronger rationale for its inclusion. You seem to think it is a negative component whereas many researchers believe fatigue is the stimulus for muscular adaptation. So to me this could be an important variable to monitor related to compliance.

You examined both hands in the study. Did you have a specific reason for doing this? Did you change the order of testing and if not could this have an impact on your findings?

Validity of the findings

In your discussion you report on the comparison of TUT between the algorithm and the two assessors but not on the agreement related to the segments of the contraction. Did you examine this aspect and if so how well did they agree? You also refer to TUT as the "gold standard" (line 388). I am not sure what you mean by this statement. Please explain.

I do appreciate you provide a lot of detail on the algorithm but must confess to not really understanding how it is actually applied by a physiotherapist or someone monitoring the exercise program. It would certainly help me, and possibly others, if you could provide more information in regard to how it can be applied in a practical setting.

Additional comments

I think this study addresses an important topic. The introduction, discussion, and conclusion seem to relate to the main purpose but a lot of the other sections seem to cover a variety of tangential areas that do not seem to relate to main purpose. The main focus of the study seems to get a bit lost and I found it quite difficult to follow and keep the main thread. However, there is a lot of good content but think you need to make it much clearer how your approach can be applied to a more practical setting which is really the main concern expressed in the introduction.

·

Basic reporting

No comment

Experimental design

No comment

Validity of the findings

No comment

Additional comments

This is an interesting and well written article. It cites adequate and relevant references. The introduction provides an adequate background. The structure, figures and tables are also adequate. However, one major problem is that the authors speculates too much about factors that they have not controlled and for which they don’t have adequate methods, like fatigue and strength. I strongly suggest removing this from the study and concentrate in the use of procedure they present. It will be necessary to expand the discussion and keep it limited to the main purpose. See below some specific comments.

Line 138 - it might be interesting to cite that previous studies reported that resistance training with elastic bands promotes similar results to resistance training performed in traditional equipaments (machines and weights) : doi: 10.1016/j.exger.2018.12.001

Line 263 - Please provide a more detailed description of the 1RM test. It is not possible to understand it and this information is very important for your article.

Line 268 - The range of load (40-60%) is very large! This imprecise estimation can influence your results regarding fadigablity, for example. Therefore, I suggest that you remove all aspects that might be influenced by not having a precise estimation of maximum and relative load and stick with your main results, that is, the validity of your procedure to monitor training and its implications.

Line 299 - Please review this phrase and consider substituting “got tired” by a more scientifical term

Line 306 – If you want to describe the subjects characteristics I recommend to do it in more detail and in a separate session (include a “Participants” topic in your methods)

Line 318-320 – I see no use in comparing participants. This is not relevant to your study and you don’t have an adequate design for that. Your number of participants is very low, your sample is heterogenous and the force measurements are not precise. With that in mind, also remover the speculation in lines 306, 346-351, the discussion from 401-450 and the conclusion from 478-484

Line 333 – The lack of standardization for the load used is a big problem, especially because you are using normalized data. According to your description the load used could be anywhere between 40-60%, therefore, the expected slope could be completely different, since higher loads will induce greater fatigue.

Line 340-342 – you cannot state that this is due to the difference in initial force because you did not measured it adequately and the you don’t have a precise measure of the relative effort. The force exerted might not mean that they are stronger, but that they applied more effort (began at a higher % of their maximum).

---

## Round 0.2 · Minor Revisions

I have received a new review of your manuscript. It concerns me that the goals of your study are still not clear enough, and it is still not clear how the participants performed the exercise. These two points are central to your work, and their clarification is mandatory. Please take into consideration the reviewer's suggestion about the description of the forearm curl.

·

Basic reporting

I think the paper is well written but the paper still needs a critical read for grammatical accuracy, spelling, and tense agreement. This seems to be particularly true of parts that have been changed or added. The authors may find it easier to report what they did and found in the past tense and not the present tense.

e.g. line 134 "they promotes.."; equipament.."
lines 273/4 "subjects were instructed to flex or extend his forearm..."

There are other parts of the manuscript that need to be corrected appreciating the fact that English is probably not the first language of the authors.

Experimental design

The study is described as a pilot study so it is really up to the journal editors to decide if they wish to publish a paper at this stage of a long term project. There are limitations due to the nature of a pilot study that the authors acknowledge.

I do feel the purpose of the study is now much clearer. The authors developed an automated system for segmenting resistance force data into contraction phase-specific
segments. Such a system could be applied to home-based resistance training programs to monitor compliance of patients. I still feel that the purpose tends to get lost a little in paper and some of the peripheral discussion.

I am still having a problem in understanding how the participants performed the exercise. Did they do it under supervised conditions or where they given the familiarization session and then instructed to perform the exercise regimen by themselves? Or are you (the authors) just concerned at this stage in seeing if the actual monitoring system is able to detect the contraction phases and not the ability of the subjects to independently follow the prescribed directions? Maybe you feel this is clear but I have read the manuscript several times and am still having problems on this point.

Validity of the findings

I think you have shown that the system you have developed is capable of showing the distinct contraction phases of an exercise such as forearm flexion. The next step would be to apply it an older demographic.

I would suggest if you are interested in an older demographic who may be confined to a wheelchair that you also investigate the application of your system to a more functional movement such as forearm extension rather than flexion as this action would seem to have more functional significance.

I also have a slight problem in describing the forearm curl as flexion and extension but not sure what to suggest in this regard. The extension phase is the eccentric loading of the forearm flexor muscles and not performed by the forearm extensors. Perhaps just describing the contraction as the concentric and eccentric components may be more explicit.

Additional comments

I appreciate you making many of the suggested changes and/or clarifying some of the parts of your study in your rebuttal.

---

## Round 0.3 · Minor Revisions

I am ready to accept your paper. I have the last question. Could you please explain to me why do you have ethical approval from a Chinese University if your experiments seem to be performed in Japan?

---

## Round 0.4 · accepted · Accept

Thank you very much for helping me to understand this situation. We can go ahead with this work.